

# The Sensed Presence Questionnaire (SenPQ): initial psychometric validation of a measure of the "Sensed Presence" experience

Joseph M. Barnby and Vaughan Bell

Department of Psychiatry, University College London, University of London, London, United Kingdom

## ABSTRACT

**Background**. The experience of 'sensed presence'—a feeling or sense that another entity, individual or being is present despite no clear sensory or perceptual evidence—is known to occur in the general population, appears more frequently in religious or spiritual contexts, and seems to be prominent in certain psychiatric or neurological conditions and may reflect specific functions of social cognition or body-image representation systems in the brain. Previous research has relied on ad-hoc measures of the experience and no specific psychometric scale to measure the experience exists to date.

**Methods**. Based on phenomenological description in the literature, we created the 16-item Sensed Presence Questionnaire (SenPQ). We recruited participants from (i) a general population sample, and; (ii) a sample including specific selection for religious affiliation, to complete the SenPQ and additional measures of well-being, schizotypy, social anxiety, social imagery, and spiritual experience. We completed an analysis to test internal reliability, the ability of the SenPQ to distinguish between religious and non-religious participants, and whether the SenPQ was specifically related to positive schizotypical experiences and social imagery. A factor analysis was also conducted to examine underlying latent variables.

**Results**. The SenPQ was found to be reliable and valid, with religious participants significantly endorsing more items than non-religious participants, and the scale showing a selective relationship with construct relevant measures. Principal components analysis indicates two potential underlying factors interpreted as reflecting 'benign' and 'malign' sensed presence experiences.

**Discussion**. The SenPQ appears to be a reliable and valid measure of sensed presence experience although further validation in neurological and psychiatric conditions is warranted.

Corresponding author
Joseph M. Barnby,
joseph.barnby@ucl.ac.uk

# INTRODUCTION

*James (1902)* first attempted to understand the sensed presence experience psychologically, describing the experience "as if there were in the human consciousness a sense of reality, a feeling of objective presence, a perception of what we may call "something there". The

philosopher and psychiatrist Karl Jaspers also discussed it in his influential book on the phenomenology of psychiatry, *General Psychopathology*, defining it as where "we are aware that something is present which at that moment is not based on any obvious sensory sign" (*Jaspers, 1913/1963*).

Contemporary researchers define the experience of sensed presence, sometimes called 'feeling of presence' or 'felt presence,' as the subjective experience of the presence of an external entity, being, or individual despite no clear sensory or perceptual evidence (*Thompson, 1982*; *Cheyne, 2001*; *Blom, 2010*; *Luhrmann, 2012*; *Luhrmann, 2013*; *Alderson-Day, 2016*). This more recent research has reported that it is particularly prevalent in certain contexts and psychological states.

One area particularly associated with the sensed presence experience is spirituality and religion. *Luhrmann & Morgain (2012)* described how participants in a prayer group frequently described the experience of a 'near tangible presence', and Luhrmann's ethnographic work (summarised in *Luhrmann, 2012*) has noted how this experience forms an essential component of evangelical religious practice. *Suedfeld & Mocellin (1987)* described the role of intense physiological states in 'spirit quests' common in many traditional religious practices that specifically induce a sensed presence experience, and *Granqvist et al. (2005)* and *Granqvist & Larsson (2006)* have demonstrated experimentally that the experience can be induced by priming participants with religious concepts.

However, the experience has also been reported in a range of other neurophysiological contexts. These include sleep-related hallucinations and paralysis, where it is typically associated with fear and anxiety (*Cheyne, Newby-Clark & Rueffer, 1999*), epileptic seizure (*Landtblom, 2006*) and particularly temporal lobe epilepsy (*Trimble & Freeman, 2006*), psychoactive drug use (*Barbosa, Giglio & Dalgalarrondo, 2005*), and direct brain stimulation (*Arzy & Schurr, 2016*); and has been associated with psychosis and auditory hallucinations (*Woods et al., 2015*), acquired brain injury (*Brugger, Regard & Landis, 1996*), Parkinson's disease (*Fénelon et al., 2011*),  and a range of intense emotional or physiological states (*Suedfeld & Mocellin, 1987*) including bereavement (*Steffen & Coyle, 2011*).

Previous theories have suggested the sensed presence phenomenon may be a result of a projected internal body map (*Brugger, Regard & Landis, 1996*), partial activation of the threat system (*Cheyne & Girard, 2007*), or a form of externalised social imagery (*Nielsen, 2007*; *Solomonova et al., 2008*), or, perhaps more exotically, an external projection of autonomous unconscious processes (*Jaynes, 2000*; *Jung, 1969*).

Notably, neuropsychological theories have been based on increasing numbers of studies where the experience has been induced in the lab, or reported in observational or patient studies, but it is noteworthy that no specific psychometric measure for the sensed presence experience exists and current studies rely on scales which are not ideally suited to the task or simple verbal description.

A seven item subscale of the Other Experiences Questionnaire (OEQ7) (Nielsen, cited in *Solomonova et al., 2008*) has been used to measure experiences akin to sensed presence experience in previous studies (*Solomonova et al., 2008*). However, the OEQ7 is actually intended to measure 'social imagery', and includes items on imaginary companions,

seeing apparitions, and the feeling of being followed, alongside items on the actual sensed presence experience.

*Trimble & Freeman (2006)* measured sensed presence in religious and non-religious individuals with epilepsy by using items from the Index of Core Spiritual Experiences (INSPIRIT) questionnaire (*Kass et al., 1991*). However, as the study used selected items from a specific spirituality questionnaire, this would not be suitable for measuring sensed presence experiences in other contexts.

Other scales include the sensed presence experience but only as a single item—such as the Tellegen Absorption Scale (*Tellegen & Atkinson, 1974*), the Magical Ideation Scale (*Eckblad & Chapman, 1983*), and the Cardiff Anomalous Perceptions Scale (*Bell, Halligan & Ellis, 2006*).

Alternatively, some studies have simply asked people to affirm whether they have had a sensed presence experience. For example, while *Hay (1979)* reported useful descriptive themes of SP experiences from participants, *Hay (1979)* and *Hay & Morisy (1978)* did not use comprehensive or validated measures to capture SP experiences and simply relied on a single question.

Given the potential for sensed presence experiences to provide a window into neuropsychological mechanisms for body representation or social cognition, clearly, a robust and validated measure of the phenomena is needed.

With this in mind, we created and investigated the reliability and validity of a new scale, called the 'Sensed Presence Questionnaire' (SenPQ), designed to capture the experience of 'sensed presence' in a psychometrically robust manner.

As religious practice has been traditionally associated with greater levels of sensed presence experience, as part of the scale validation we predicted that individuals who have religious practice / belief from the general population would score higher on the SenPQ as people without. Based on previous research, we also predicted that the SenPQ would selectively correlate with measures of unusual perceptual experiences but no other aspects of schizotypy, as well as correlating with measures of social imagery and daily spiritual experience.

## MATERIALS AND METHODS

A cross-sectional observational design was used in the general population. Data was collected in the form of an online survey using two distinct samples. The study was reviewed and ethically approved by the UCL ethics review board (ref no.: 8587/001). Participants indicated consent on the online form.

### Design of the Sensed Presence Questionnaire (SenPQ)

The Sensed Presence Questionnaire (SenPQ) is comprised of 16 questions. These were derived from a literature review of the sensed presence phenomenon spanning studies from sleep paralysis, epilepsy and other neurological disorders, psychosis, stress and anxiety, ritual, drug induced experiences, and the general population. As well as covering a range of typical sensed presence experiences from the scientific literature, the scale also includes

items that are positively and negatively valenced, as well as neutrally valenced in their presentation.

The questionnaire requests that respondents refer to experiences from the last month only when rating the items, and not to record any perceptions associated with drug-induced experiences. Respondents are asked to indicate the frequency with which the experience has occurred using a Likert-like scale: 'Never', 'Occasionally', 'Sometimes', 'Very Often', 'Always'. The questionnaire is freely available online and has been released under a Creative Commons license at the following link: https://osf.io/fecgz/.

### Participants

Participants were recruited via two methods: (i) online via the http://proflic.ac online study recruitment platform that has diverse participant base and where we received 101 completed responses from separate individuals (Sample S1) from 135 responses in total including incomplete responses. In addition, social media advertisements were sent from the authors' personal accounts and accounts associated with the authors' university department (with notices that did not refer to anything spiritual or religious) and emails were sent to religious groups including university religious societies (Hindu society, Islamic Society, Christian Union, Sikh Society, Buddhist Society, and the Jewish Society) and local churches requesting participants (Sample S2). Recruitment for both samples was started in parallel. The questionnaire took approximately 25 min to complete. Participants recruited via the online recruitment service were paid £4 upon questionnaire completion. All participants were directed to the same online questionnaires.

Individuals who clicked on the link to the survey and began to fill in the survey were considered to have been recruited into the study, regardless of whether the scales were fully completed, although only complete questionnaire sets were entered into the analysis. Questionnaires were preceded by a page requesting demographic data that required age, gender, religious belief, average meditation practice, ethnic group, and education level.

Both samples were combined for the final analysis which consisted of a total of 191 participants included (see Fig. 1). Analysis scripts that conduct a separate analysis for each sample (minus the principal components analysis which is underpowered when not conducted on the whole sample) are available on the Open Science Framework page for this study (https://osf.io/fecgz/), which demonstrate the same pattern of results in each sample, supporting the reliability and validity of the scale.

### Additional measures

Other Experiences Questionnaire—social imagery subscale (OEQ7) (*Solomonova et al., 2008*): a validated subscale designed to capture social imagery, and has previously been established as reliable and valid in the context of sleep paralysis and anxiety. The frequency of each item is rated on a 4-point Likert scale. This measure has been previously used to measure an aspect of the sensed presence experience, and we predicted that it would correlate with SenPQ scores, indicating convergent validity. The internal reliability for the scale in this study, measured using Cronbach's Alpha, was 0.706.

Brief Oxford-Liverpool Inventory of Feeling and Experiences (O-LIFE) (*Mason, Linney & Claridge, 2005*): a briefer, validated version of the O-LIFE schizotypy scale (*Claridge*

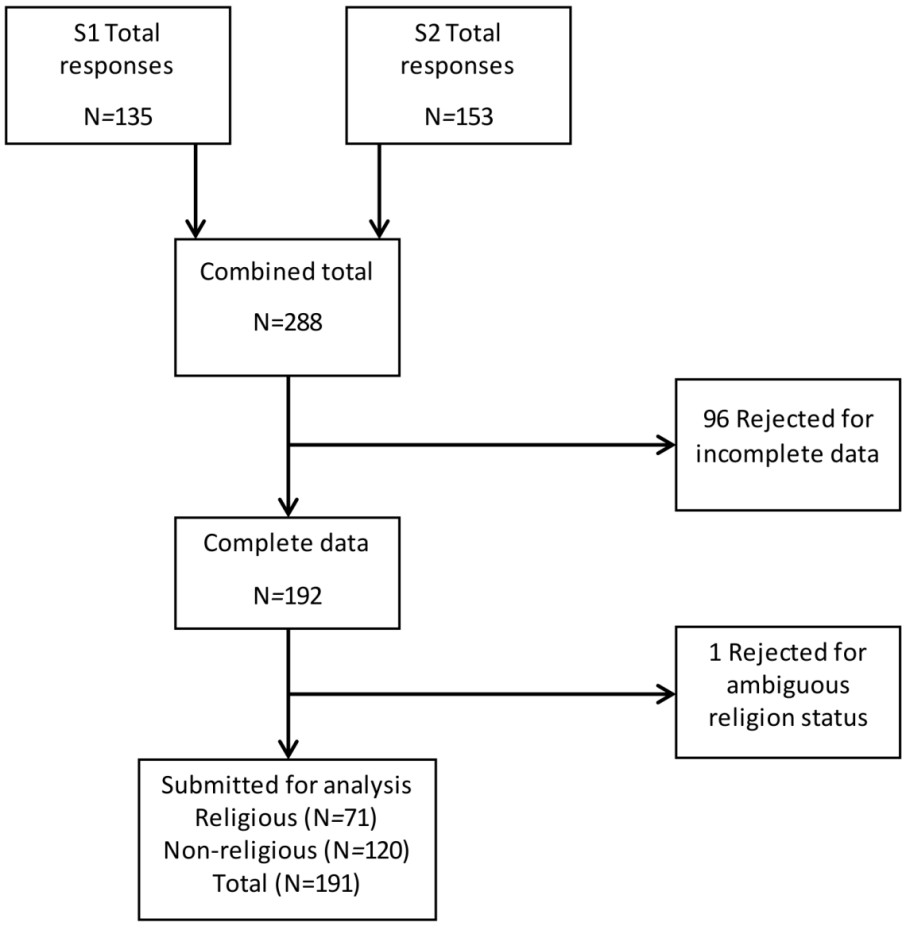

**Figure 1  Recruitment diagram.**

*et al., 1996*) that includes four subscales that measure different aspects of schizotypy: unusual experiences (UE), cognitive disorganisation (CD), introvertive anhedonia (IA), and impulsive non-conformity (IN). Each item is rated on a binary scale, with participants either affirming or disagreeing with each statement. We predicted a selective association between the SenPQ and the unusual experiences subscale of the O-LIFE, indicating that the sensed presence experience reflected a form of reality distortion experience but no other cognitive or behavioural aspects of schizotypy. Internal reliability of the scale in this study was 0.863.

Daily Spiritual Experience Scale (DSES) (*Underwood & Teresi, 2002*) is originally a 16-item validated scale designed to assess spiritual experiences in daily life. This scale was based on qualitative research and has been tested for reliability and validity in a range of populations (*Underwood, 2011*). The first 15 items are answered on a 6-point Likert scale that ask about frequency of particular spiritual experiences from 'many times a day' to 'never or almost never'. The final item is a general question about 'how close do you feel to God' and is answered on a four point Likert scale from 'not close at all' to 'as close as possible'. This item was omitted as it has been found not to be reliable for some Hindu

and Muslim respondents (*Husain et al., 2016*) who were groups we specifically invited to participate. However, the DSES has been shown to be reliable and valid in this 15-item form *Kim, Martin & Nolty, (2016)* and *Kurian et al. (2016)*. Considering that the sensed presence experience has been reportedly more frequently in spiritual and religious contexts, we predicted a positive correlation with the SenPQ. Internal reliability of the scale in this study was 0.936.

We also included two scales that measured general psychological well-being and social interaction anxiety for an exploratory analysis and use as potential covariates, given that sensed presence experience has been previously associated with psychopathology in some populations.

WHO-5 Well-Being Index (*World Health Organization, 1998*): a brief, validated well-being questionnaire, shown to capture well-being as accurately as its extended version (*Topp et al. , 2015*). Each item describes an aspect of well-being and the respondent is asked rate how present it has been during the last two on a 6-point Likert scale ranging from 'all of the time' to 'at no time'. Internal reliability of the scale in this study was 0.844.

The Social Interaction Anxiety Scale (SIAS) (*Mattick & Clarke, 1998*): a validated scale designed to capture anxiety associated with meeting, conversing, or mixing with others. Each item describes anxiety associated with a particular social interaction scenario and the respondent is asked to rate each item on a 5-point Likert scale from 'not at all' to 'extremely'. Internal reliability of the scale in this study was 0.950.

Participants were also given the option of a free text box at the end of the questionnaire series to describe a sensed presence experience in their own words if they indicated they had experienced one. This information was not used in the current study.

## Analysis

Internal reliability of the SenPQ was tested with Cronbach's Alpha (*Cronbach, 1951*). Validity was determined firstly by whether SenPQ score distinguished between religious and non-religious groups by using an independent samples $t$-test, and we predicted a significant difference between SenPQ in religious and non-religious groups, indicating discriminant validity. Secondly we examined the association between SenPQ score and additional measures, predicting that the SenPQ would selectively correlate with the O-LIFE unusual experiences subscale, the OEQ7 measure of social imagery and the DSES measure of spiritual experience, indicating convergent and divergent validity. We had no specific predictions about associations with the measures of anxiety and well-being, as sensed presence experiences have been previously associated with a range of affective states (*Alderson-Day, 2016*).

In addition, we completed an exploratory principal components analysis of the SenPQ items on the entire sample using direct oblimin rotation based on the assumption that underlying factors would not be independent. The mean item scores for each individual factor loading were used for all subsequent analysis. Parallel analysis for principle components (95% confidence interval, 1,000 random correlation matrices) (*Horn, 1965*; *O'Connor, 2000*) and observation of the scree plot (*Cattell, 1966*) were used to select

retained factors. Subsequent factor analyses were completed to specifically extract models for specific numbers of retained factors.

An Item Response Theory (IRT) analysis was performed on the entire dataset for scores on the SenPQ. IRT is a model-based theory that measures the responses between items and the trait that each item is supposedly measuring (*Emberston & Reise, 2000*)—in this case, the sensitivity to sensed presence experiences. A Graded Response Model (GRM; *Samejima, 1969*) was used in this particular IRT analysis—this is due to the SenPQ using a polytomous scoring system. The GRM model attributes each item as a series of response dichotomies, or thresholds ($\beta$) compared to the discrimination parameter, or slope ($\alpha$)—because the SenPQ items comprised 5 graded choices (1- Never to 5- Always) this represents four $\beta$: 1 vs 2–5 ($\beta_1$), 1–2 vs. 3–5 ($\beta_2$), 1–3 vs. 4–5 ($\beta_3$), and 1–4 vs. 5 ($\beta_4$). These thresholds correspond to the trait level ($\theta$) at which a new individual answering the questionnaire has a 50% chance of endorsing the relevant or higher response. These thresholds are not the same for each item, and each item will have its own set of thresholds.

Graphical illustrations of this relationship were also created. An Item Information Function (IIF), Test Information Function (TIF), and Test Characteristic Curve (TCC) was generated for the SenPQ. An IIF displays at which locations along the trait scale each item is able to be accurate about a new individual using the questionnaire. A TIF reports the level of precision of the entire measure at different points along the trait scale. A TCC shows the expected score of the measure at different points along the trait scale.

We also completed a receiver operating characteristic (ROC) analysis to examine how specific scale scores distinguished between religious and non-religious groups to additionally test discriminant validity across the range of the measure.

## RESULTS

### Demographics

The total sample consisted of 89 males, 99 females, 2 who selected 'Other' and 1 who selected 'Prefer not to say'. The mean age of the sample was 36.23 (SD = 13.4; Range 17–73). Religious affiliation, ethnicity and level of education are reported in Table 1. For the purposes of classifying people into religious and non-religious groups for further analysis, people who selected 'No Religion' or 'Agnostic' were considered non-religious, as were individuals who recorded themselves as 'Atheist' under the 'Other' option. All others were considered religious.

20 participants reported daily meditation practice, 10 weekly practice, 23 monthly practice, and 138 no practice. Self-reported ethnicity of the sample was "English / Welsh / Scottish / Northern Irish / British" ($N = 131$), "White other" ($N = 25$), "Indian" ($N = 8$), "White and Asian" ($N = 5$), "Chinese" ($N = 4$), "White and Black African" ($N = 3$), "Pakistani" ($N = 2$), "Other" ($N = 2$), "Irish" ($N = 2$), "White and Black Caribbean" ($N = 2$), "Other Mixed / Multiple ethnic background" ($N = 2$), "Bangladeshi" ($N = 1$), "African" ($N = 1$), "Caribbean" ($N = 1$), "Arab" ($N = 1$), "Gypsy or Irish Traveller" ($N = 1$).

**Table 1 Religion and education of sample.**

|  | Frequency (%) |
|---|---|
| Religious affiliation |  |
| No religion | 99 (51.83) |
| Agnostic | 16 (8.37) |
| Christian | 45 (23.56) |
| Buddhist | 4 (2.09) |
| Hindu | 2 (1.04) |
| Jewish | 3 (1.57) |
| Muslim | 3 (1.57) |
| Sikh | 3 (1.57) |
| Other | 16 (8.37) |
| Highest level of education |  |
| GCSE | 10 (5.23) |
| A level | 29 (15.18) |
| University undergraduate | 93 (48.69) |
| University postgraduate | 59 (30.89) |

## Statistical analysis

Due to the sampling distribution of mean scores on the SenPQ violating the assumption of normal distribution, all analyses were conducted using a simple bootstrap re-sampling method (1,000 samples, 95% CI) (*Bland & Altman, 2015*). All data analysis used SPSS v.22 (SPSS Inc.), except for the IRT analysis which used STATA v.14 (Stata Corp). The raw data and analysis scripts for this study are freely available online at the Open Science Framework at the following link: https://osf.io/fecgz/.

## Internal reliability

All SenPQ items where entered into internal reliability analysis and the scale demonstrated very high internal consistency (Cronbach's alpha = 0.951).

## Validity

Means and standard deviations for the scale scores are displayed in Table 2. Discriminant validity of the SenPQ was demonstrated by conducting an independent samples $t$-test (two-tailed) between mean scores from religious ($N = 71$) and non-religious groups ($N = 120$). The religious group had a higher mean score than the non-religious group (see Table 2), a difference which was significant when tested with an independent samples $t$-test ($t = -3.592$, $p = 0.002$, mean difference $= -5.208$, 95% CI [$-8.098$–$-2.156$]; Cohen's $d = 0.51$), indicating good discriminant validity.

As can be seen in Table 3, the SenPQ demonstrated a strong significant correlation with the OEQ-7 social imagery scale and a moderate significant correlation with the DSES daily spiritual experiences scale. There was a strong significant correlation with the unusual experiences subscale of the O-LIFE schizotypy scale, a weak correlation with the impulsive non-conformity subscale, and no significant correlation with the cognitive disorganisation or introvertive anhedonia subscales, indicating good convergent and divergent validity.

**Table 2  Descriptive statistics for the religious, non-religious, and total samples.**

| Group | N | Age | Gender (M:F:O) | SenPQ | BSenPQ | MSenPQ | OEQ7 | O-LIFE Total | UE | CD | IA | IN | DSES | WHO-5 | SIAS |
|---|---|---|---|---|---|---|---|---|---|---|---|---|---|---|---|
| Religious | 71 | 36.20 (13.04) | 35:35:1 | 26.55 (11.86) | 15.35 (7.43) | 12.93 (5.67) | 10.85 (2.94) | 14.42 (7.73) | 4.14 (3.20) | 4.24 (2.83) | 2.80 (1.98) | 3.24 (2.08) | 44.93 (15.60) | 19.97 (4.29) | 46.13 (14.22) |
| Non-Religious | 120 | 36.25 (13.66) | 54:64:2 | 21.34 (8.14) | 11.57 (4.86) | 11.17 (4.44) | 9.34 (2.65) | 14.47 (7.47) | 3.12 (2.77) | 4.62 (2.96) | 3.22 (2.39) | 3.51 (2.33) | 30.28 (11.48) | 19.34 (4.58) | 49.59 (17.83) |
| Total | 191 | 36.23 (13.40) | 89:99:3 | 23.28 (9.98) | 12.97 (6.21) | 11.82 (5) | 9.90 (2.84) | 14.45 (7.55) | 3.50 (2.97) | 4.48 (2.91) | 3.07 (2.25) | 3.41 (2.24) | 35.72 (14.92) | 19.58 (4.48) | 48.30 (16.62) |

**Notes.**

SenPQ, Sensed Presence Questionnaire; BSenPQ, Benign Sensed Presence Questionnaire factor items; MSenPQ, Malign Sensed Presence Questionnaire factor items; O-LIFE, Brief Oxford-Liverpool Inventory of Feelings and Experiences; UE, Unusual Experiences subset; CD, Cognitive Disorganisation subset; IA, Introvertive Anhedonia subset; IN, Impulsive Non-Conformity subset; OEQ-7, Other Experiences Questionnaire; WHO-5, World Health Organisation 5-item well-being questionnaire; DSES, Daily Spiritual Experience Scale; SIAS, Social Interaction Anxiety Scale.

**Table 3  Pearson correlations between SenPQ and other scales in the total sample.**

| | O-LIFE | | | | | | | | |
| | UE | CD | IA | IN | Total | OEQ-7 | WHO-5 | DSES | SIAS |
|---|---|---|---|---|---|---|---|---|---|
| SenPQ | 0.641[***] | 0.110 | 0.068 | 0.308[***] | 0.406[***] | 0.673[***] | 0.056 | 0.407[***] | 0.025 |

**Notes.**

SenPQ, Sensed Presence Questionnaire; O-LIFE, brief Oxford-Liverpool Inventory of Feelings and Experiences; UE, O-LIFE Unusual Experiences subscale; CD, O-LIFE Cognitive Disorganisation subscale; IA, O-LIFE Introvertive Anhedonia subscale; O-LIFE IN, Impulsive Non-Conformity subscale; OEQ-7, Other Experiences Questionnaire; WHO-5, World Health Organisation 5-item well-being questionnaire; DSES, Daily Spiritual Experience Scale; SIAS, Social Interaction Anxiety Scale.

[***] $p < 0.001$

## Demographic analysis

No significant correlation between the age and SenPQ score (Pearson $r = 0.026$, $p = 0.723$) was found and no significant difference between genders when tested with an independent samples $t$-test ($t = 1.268$, $p = 0.206$). There was no significant effect of education level on SenPQ score when tested with a one-way between subject ANOVA ($F_{(3, 187)} = 1.100$, $p = 0.350$). Because so few people reported meditation practice at the more frequent end of practice, this was collapsed into a binary 'yes/no' variable. People who reported engaging with any sort of meditation practice were significantly more likely to score higher on the SenPQ when tested with an independent samples $t$-test ($t = 3.222$, $p = 0.001$).

## Measures of anxiety and well-being

No significant associations were found between the SenPQ and WHO-5 score ($r = .56$, $p = .443$) and between the SenPQ and SIAS score ($r = .025$, $p = .773$).

## Factor analysis of the SenPQ

To investigate the factor loadings of the Sensed Presence Questionnaire, a factor analysis was run on all 16 items.

To test assumptions, a Kaiser–Meyer–Olkin measure of sampling adequacy (0.925) and Bartlett's test of sphericity ($\chi 2(120) = 2467.009$, $p < 0.001$) were run and considered adequate, with all items significantly correlating by at least 0.3 ($p < 0.001$).

An initial factor analysis suggested a two factor solution based on inspection of the scree plot that indicated a clear break after two components (*Cattell, 1966*). The parallel analysis conducted using *O'Connor*'s (*2000*) method suggested a one factor solution, based on the fact that only one eigenvalue from the study data set was greater than the simulated equivalent for randomly generated correlation matrices, with the second factor being marginally below the cut-off. We subsequently conducted two separate factor analyses that specifically extracted one and two component solutions and subsequently judged the two component solution to be more interpretable. The first factor was interpreted as 'benign presence' and the second factor was interpreted as 'malign presence'. The first component explained 52.28% of the variance, and the second component explained an additional 7.66% of the variance in the sample. The pattern matrix can be found in Table 4.

**Table 4 Pattern matrix factor loadings from two-component exploratory factor analysis of item scores from 191 samples.** All loadings less than 0.4 are not displayed.

| | Item | Factor 1 | Factor 2 |
|---|---|---|---|
| 13 | I have felt the presence of a protective being around me that I couldn't see | 0.990 | |
| 6 | I have felt I was being watched over by caring being that I couldn't see | 0.981 | |
| 3 | When I was under a lot of pressure, I felt someone or something was accompanying me | 0.823 | |
| 8 | I have felt when an unseen presence has arrived | 0.707 | |
| 5 | During times of stress I have had the feeling that I was being accompanied by an unseen presence | 0.604 | |
| 11 | I have visited certain places where I can feel the presence of distinct but unseen beings | 0.581 | |
| 12 | I can feel the presence of people that I know are physically distant from me | 0.559 | |
| 15 | Even though I knew the person had died, I felt them accompanying me | 0.550 | |
| 7 | When I have visited specific locations, I felt I was in the presence of an unseen being or beings | 0.494 | 0.432 |
| 1 | I have felt another being or beings near me when I couldn't see anyone around me that could explain it | | 0.544 |
| 10 | I have felt as if someone or something is near me, even though I know it is not really the case | | 0.578 |
| 2 | When half asleep I have thought someone else was with me, only to find out when I woke up that they couldn't have been | | 0.693 |
| 16 | Even though I knew it was my imagination, I still felt as if someone or something was with me | | 0.701 |
| 9 | I have woken up during the night with the feeling that an unseen presence was in the room with me | | 0.702 |
| 14 | I have felt a sinister or threatening presence around me, despite not being able to see any evidence for it | | 0.822 |
| 4 | I have had the feeling that a negative or hurtful presence was around me that I couldn't see | | 0.896 |

## Item-response theory graded response modelling

The SenPQ was submitted to an IRT GRM bootstrap analysis (1,000 replications) to understand the relationship between difficulty of items and sensitivity to sensed presence experiences.

The analysis reported that all items had moderate to large slopes ($\alpha = 1.76$–$4.51$) with a model log likelihood value of $-1947.92$. All $\beta$ values were evenly spaced and ascending from $\beta_1$ to $\beta_4$. Two $\beta_4$ thresholds were not available because no participant answered '5-Always' on item 3 and 7: 'When I was under a lot of pressure, I felt someone or something was accompanying me', and 'When I have visited specific locations, I felt I was in the presence of an unseen being or beings', respectively.

Table 5 shows the results for the slopes ($\alpha$) and threshold parameters ($\beta1$–$\beta4$) for all items on the SenPQ.

**Table 5 Item response theory item-parameter estimates for all 16 items on the SenPQ.**

| Items | α | β₁ | β₂ | β₃ | β₄ |
|---|---|---|---|---|---|
| 1 | 3.12 (0.52) | 0.61 (0.11) | 1.56 (0.15) | 2.22 (0.21) | 2.88 (0.25) |
| 2 | 1.97 (0.29) | 0.36 (0.12) | 1.32 (0.16) | 2.31 (0.29) | 3.51 (0.41) |
| 3 | 3.40 (0.57) | 0.71 (0.1) | 1.40 (0.14) | 1.97 (0.18) | * |
| 4 | 2.45 (0.33) | 0.92 (0.14) | 1.67 (0.18) | 2.61 (0.27) | 3.09 (0.34) |
| 5 | 4.51 (0.98) | 0.71 (0.11) | 1.26 (0.13) | 1.83 (0.16) | 2.56 (0.23) |
| 6 | 2.75 (0.42) | 0.60 (0.11) | 1.33 (0.13) | 2.01 (0.18) | 2.72 (0.26) |
| 7 | 3.81 (0.55) | 0.51 (0.1) | 1.14 (0.12) | 1.80 (0.13) | * |
| 8 | 2.93 (0.45) | 0.95 (0.13) | 1.51 (0.15) | 1.89 (0.19) | 2.47 (0.28) |
| 9 | 1.76 (0.32) | 0.54 (0.14) | 1.42 (0.20) | 2.50 (0.40) | 3.74 (0.54) |
| 10 | 3.14 (0.48) | 0.35 (0.09) | 1.26 (0.15) | 2.17 (0.2) | 2.89 (0.23) |
| 11 | 4.36 (0.69) | 0.56 (0.1) | 1.31 (0.12) | 1.85 (0.15) | 2.64 (0.2) |
| 12 | 2.13 (0.39) | 0.86 (0.15) | 1.60 (0.20) | 2.19 (0.24) | 2.87 (0.33) |
| 13 | 3.33 (0.56) | 0.75 (0.12) | 1.37 (0.12) | 1.89 (0.15) | 2.38 (0.23) |
| 14 | 2.12 (0.38) | 0.97 (0.14) | 1.94 (0.25) | 2.52 (0.31) | 3.04 (0.38) |
| 15 | 2.31 (0.38) | 0.78 (0.12) | 1.52 (0.19) | 2.29 (0.31) | 3.23 (0.38) |
| 16 | 2.91 (0.46) | 0.31 (0.10) | 1.29 (0.14) | 2.14 (0.20) | 2.89 (0.25) |

**Notes.**

α, discrimination parameter (slope); β₁–β₄, threshold parameters (residuals).

All numbers in brackets are the bootstrapped standard error values.

*Residual unable to be calculated.

Observing the slopes and the residuals suggests that all items were providing high item level and test level information, with item 9 being the 'easiest', and item 5 the 'hardest'.

Residuals at the test and item level were also analysed graphically. Figure 2 displays the IIF, TIF, and TCC for the SenPQ.

Because a large portion of the curves are above $0\theta$ in Fig. 2, this suggests that the SenPQ is better designed for respondents with higher sensitivity to sensed presence experiences.

## ROC analysis

Data from the psychometric measures was entered into a ROC analysis to show they discriminated religious from non-religious groups over the extent of their score range. Results are displayed in Table 6 and Fig. 3.

The most efficient total discriminator of religious and non-religious groups was the DSES. The SenPQ and OEQ7 were highly discriminant and perform almost identically. The Unusual Experiences subscale of the O-LIFE schizotypy scale discriminates between groups to a lesser extent, and all other scales show no significant discriminant ability.

## DISCUSSION

This study involved the creation and initial validation of a 16-item sensed presence questionnaire. In a general population sample, we demonstrated that the SenPQ is a reliable and valid measure for measuring the experience of 'sensed presence'.

One of the clearest findings is that the experience of 'sensed presence' is quite common in the general population, even among those who profess no religious affiliation. However, the

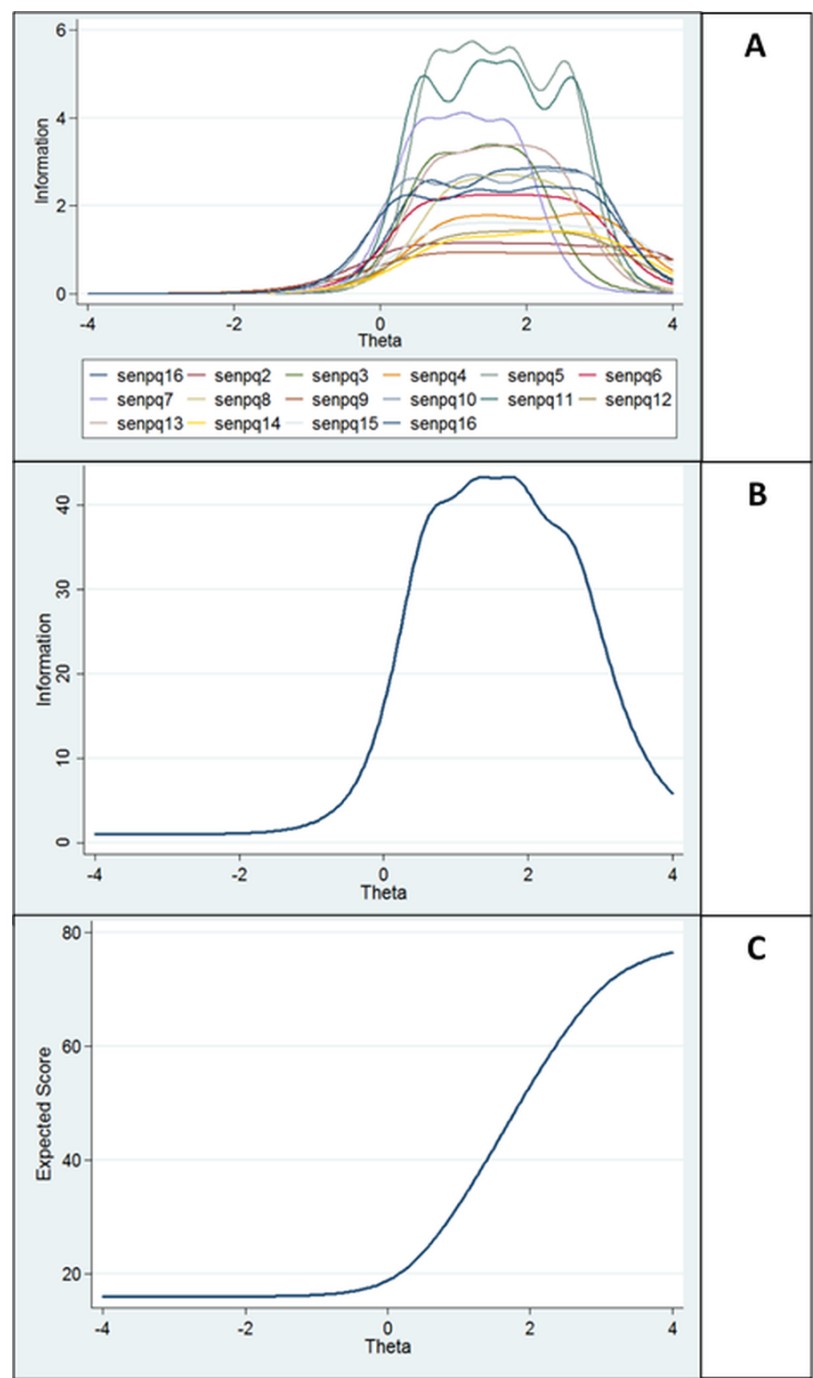

**Figure 2** **Item response theory graphical analyses at the test and item level of the SenPQ.** Item information function (A), Test information function (B), and Test characteristic curve (C).

**Table 6** Receiver operator characteristic analysis of all measures predicting religious (positive) or non-religious (negative) identification.

| Measure | Area under the curve | Std. error | 95% CIs | |
| --- | --- | --- | --- | --- |
| SenPQ | .655*** | .041 | .575 | .735 |
| UE | .592* | .042 | .509 | .675 |
| CD | .462 | .043 | .379 | .546 |
| IA | .458 | .042 | .376 | .540 |
| IN | .473 | .043 | .390 | .556 |
| O-LIFE | .490 | .043 | .406 | .574 |
| DSES | .811*** | .034 | .745 | .878 |
| OEQ7 | .656*** | .041 | .576 | .735 |
| WHO-5 | .547 | .043 | .464 | .630 |
| SIAS | .459 | .042 | .377 | .541 |

**Notes.**

SenPQ, Sensed Presence Questionnaire; O-LIFE, brief Oxford-Liverpool Inventory of Feelings and Experiences; UE, Unusual Experiences O-LIFE subscale; CD, Cognitive Disorganisation O-LIFE subscale; IA, Introvertive Anhedonia O-LIFE subscale; IN, Impulsive Non-Conformity O-LIFE subscale; OEQ-7, Other Experiences Questionnaire; WHO-5, World Health Organisation 5-item well-being questionnaire; DSES, Daily Spiritual Experience Scale; SIAS, Social Interaction Anxiety Scale.

Asymptotic significance: $*p < 0.05$, $***p < 0.001$.

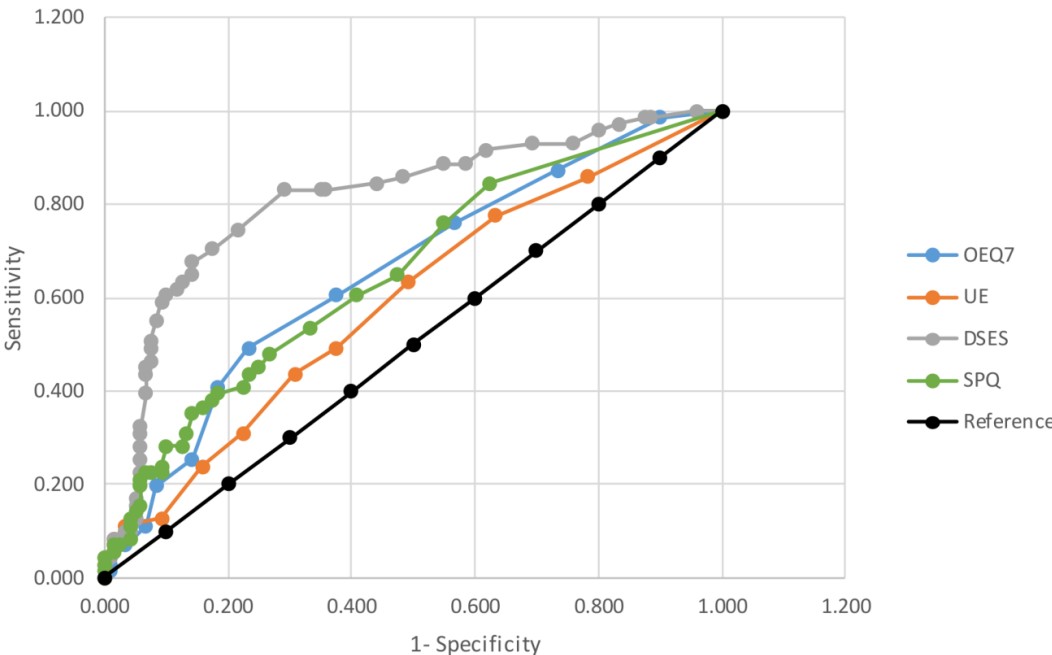

**Figure 3** Graph of receiver operating charactistic (ROC) curve for DSES, OEQ7, UE, and SenPQ compared to the null reference in classifying religious and non-religious participants. SenPQ, Sensed Presence Questionnaire; DSES, Daily Spiritual Experience Scale; OEQ-7, Other Experiences Questionnaire; UE, Unusual Experiences O-LIFE subscale.

experience was more commonly reported by people who either classify themselves as having a specific religion or who score more highly on a measure of daily spiritual experience, as was predicted from previous research (*Luhrmann & Morgain, 2012*; *Luhrmann, 2012*; *Granqvist et al., 2005*; *Granqvist & Larsson, 2006*).

However, considering that the experience of sensed presence has previously been reported in the context of various states and conditions outside of a religious framework, including sleep disorders, neurological conditions, drug use, and intense physiological and emotional stress, we hope that the Sensed Presence Questionnaire to be useful across a range of presentations and this needs to be a focus for future validation studies.

Scores on the Sensed Presence Questionnaire showed a strong association not only with social imagery and spiritual experience but also, selectively, with the unusual experience subscale of the O-LIFE schizotypy scale, suggesting a link with hallucinatory and magical thinking experiences on the psychosis continuum (*Mason & Claridge, 2006*). It has been argued previously (*Bell et al., 2017*) that the positive symptoms of psychosis involve, at least in part, the atypical activation of social cognitive systems for representing others, and we hypothesis that the sensed presence experience may represent a state of minimal social agent representation. Notably, the association with the O-LIFE scales was selective and there was no marked relationship between sensed presence experience and other aspects of schizotypy that don't represent hallucinatory experience. The fact that there was no association with social anxiety or general well-being may suggest, measurement error aside, that the sensed presence experience may reflect a form of minimal social agent representation which is heightened in people who have higher levels of hallucinatory experience and is not just social anxiety-related hypervigilance.

Two factors emerged from the factor analysis that were interpreted as 'benign' and 'malign', echoing reports from the literature on differing emotional valence of sensed presence experiences (*Alderson-Day, 2016*). Although seemingly a good conceptual fit to previous reports, it is worth sounding a note of caution. Firstly, the factor analyses were exploratory and a confirmatory factor analysis needs to be conducted on an independent sample before the concept of positively and negatively valenced sensed presence experiences as distinct latent variables in the general population can be accepted with confidence. Furthermore, the study was conducted with a sample where members of religious groups were specifically invited to allow for a strong comparison. We are aware that this may have over-represented people with benign sensed presence experiences and a more representative sample of the general population is needed to be sure the factor structure can be generalised. It is also likely that in people with associated medical conditions, sensed presence experiences may arise from an impact on specific social neurocognitive process and these may be quite different in terms of structure.

Although our sample was diverse in terms of ethnicity, religion, and age, it also over-sampled people with higher levels of education. Considering this, further validation needs to be conducted using methods that are more likely to yield samples that are representative of the general population.

Additionally, questionnaires were presented to the sample in a fixed order and it is possible that order effects may have had an influence on responding via response bias. It

is also worth bearing in mind that when enquiring about experiences related to mental health and religion, social desirability-related response biases need to be considered. While we assume that the paradigm used here, an anonymously completed online study, would be among the least subject to direct social desirability bias, we are aware that these areas have strong social stereotypes attached to them and internalised biases may be potential influences. Future studies could balance presentation order and use social desirability measures to exclude or adjust for these possible effects.

## CONCLUSIONS

From the data presented here, the Sensed Presence Questionnaire (SenPQ) appears a reliable and valid measure of the 'sensed presence' experience. Initial principal components analyses suggest that the SenPQ may comprise of two factors, malign and benign presence. We hope the scale will be subject to further validation studies and will allow the 'sensed presence' experience to be investigated in a range of conditions.

### Funding
The authors received no funding for this work.

### Competing Interests
The authors declare there are no competing interests.

### Author Contributions
- Joseph M. Barnby conceived and designed the experiments, performed the experiments, analyzed the data, wrote the paper, prepared figures and/or tables, reviewed drafts of the paper.
- Vaughan Bell conceived and designed the experiments, analyzed the data, wrote the paper, prepared figures and/or tables, reviewed drafts of the paper.

### Human Ethics
The following information was supplied relating to ethical approvals (i.e., approving body and any reference numbers):

The University College London Ethics Committee granted ethical approval to carry out this study in the general population (Ethics ref no.: 8587/001).

### Data Availability
osf.io (Open Science Framework)

Sensed Presence Questionnaire Validation Study

Identifiers: DOI http://dx.doi.org/10.17605/OSF.IO/FECGZ | https://osf.io/fecgz/.

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
