# Peer review of "The Sensed Presence Questionnaire (SenPQ): initial psychometric validation of a measure of the “Sensed Presence” experience"

_PeerJ, doi:10.7717/peerj.3149_

## Round 0.1 · original submission · Major Revisions

Please address all of the reviewers' comments completely. Although one of the reviewers was concerned about the propriety of the topic for PeerJ, I feel that the study lies within PeerJ's purview.

Reviewer 1 ·

Basic reporting

The paper is written in unambiguous and professional English. The only remarks in this sense are referred to some typos here and there in the paper. Therefore I suggest a careful reading of the manuscript in order to identify and fix all of these typos.

The paper provides an adequate literature exploration.

I am not an expert in the field of Sensed Presence since my research interests are focused on psychometrics and mathematical psychology therefore I will focus my comments on these aspects of the paper.

Experimental design

As I said, I am not an expert in the field of Sensed Presence, nevertheless I do not feel that this topic fits the aims and scope of the journal. Of course, the editor could have a different opinion on this.

The methods of the research are accurately described and the procedure reflects good technical and ethical standards.

Validity of the findings

In my opinion the topic of the paper would have a low impact in the field.

Nevertheless the data collection is adequate (even if some shortcomings in the sampling are present).

I would suggest to apply an IRT analysis. Moreover, in my view, this kind of research should contain enough data to allow for a two step factor analysis on two independent samples: an exploratory and a confirmatory factor analysis.

Without such analysis, results are too weak. In the case at hand, it would be necessary to confirm the goodness of the factorial structure since the second factor accounts for a very low amount of variance, especially if compared to the first one. A confirmatory analysis which compares the mono factorial solution with a two factor model should be conducted.

Additional comments

In my opinion the paper is potentially interesting and the analysis conducted so far are adequate. My feeling is that what has been done is ok, but it is not sufficient for the aim of the paper.

·

Basic reporting

This is an interesting and well-written paper that discusses topics of relevance to the study of unusual perceptions and experiences. The authors give an important contribution to the study of "sensed presence" experience by developing and validating an original measure of this construct. The introduction offers a concise but reasonable rationale for the study and does a good job in arguing for the need of reliable instruments to measure these and other similar experiences. The results are interesting and the paper provides a clear presentation of the data. The SenPQ proved to be reliable and valid, but comments are made (below) on potential problems concerning item construction and interpretation. Additional analyses are suggested, as well as consideration of alternative hypotheses for the findings. In summary, this paper is certainly worthy of publication once some relatively minor issues are dealt with.

Experimental design

The methodological approach is sound and the statistical analyses are adequate. The data collection followed strict ethical standards and the research question and purposes are meaningful.

1) However, the justification for the selection of the additional measures is not entirely clear, and is merely deduced from the introduction. With the exception of the SenPQ, the presentation / description of the other measures strikes me as far too short and should be improved and expanded. Considering the relevance of these measures for the (convergent) validity of the SenPQ, It would be relevant to provide some information on their psychometric characteristics and also on their specific contributions for the investigation of the study's hypotheses concerning "sensed presence" experiences.

2) I suggest that you improve the description of the Daily Spiritual Experiences Scale at lines 198-200. It is not clear why question 16 was omitted from your version of the scale, and the mere inclusion of Ellison & Fan (2008) reference is not in itself sufficient as an explanation.

3) Another important issue concerns the interpretation of item responses. For example, in question 6 ("I have felt I was being watched over by caring being that I couldn't see"), participants could answer positively not because of any sensed presence experience, but because of their belief in the intervention of God (or other spiritual being) in human life. Thus, the item could eventually be interpreted metaphorically (maybe conveying a symbolic meaning), rather than literally. Future investigations on the validity of the SenPQ could ask participants on their specific interpretations of each item, what could help improve the questionnaire.

4) At line 100, you seem to classify the "feeling of being followed" as a different experience from the actual sensed presence experience. But are these two experiences different in any sense? Why not consider including the feeling of being followed and the sense of being stared at as examples or variations of the sensed presence experience? Although some items in the questionnaire could eventually be interpreted by respondents as implying these other experiences, it would be relevant to consider the inclusion of items specifically related to the feeling of being followed, in order to assess their relationship with the other questionnaire items.

Validity of the findings

The results are interesting and the conclusions are well stated.

1) Notwithstanding, a detailed discussion of the correlations between the SenPQ and each additional measure would be appropriate. Links with the hypotheses covered in the introduction section should also be provided.

2) Considering that the factor analysis indicated a two-factor solution as the most adequate, it would be relevant to include participants' mean score on each subscale in table 2, and not only their total score. Data on the parallel analysis results would also strength the presentation of the findings.

3) The study included a series of relevant information on the participants' demographics, including age, gender, religious affiliation, meditation practice and many other. However, no data was offered on the relationship between SenPQ and the demographic variables. When checking the data available on the Open Science Framework, I found no gender and age differences for the SenPQ total and subscale scores. However, this does not mean that these results should be overlooked. For example, they seem to contradict the literature indicating women as more prevalent among those reporting religious / spiritual or paranormal experiences. Did you find any evidence of a relationship between SenPQ and meditation practice? This information is also of relevance, considering the possible influence of religious / spiritual practice on reports of unusual experiences.

4) The results showed no association of sensed presence with well-being and social anxiety. One possible explanation for these findings is the lack pf empirical association between these variables. But another possibility is response bias. Factors such as response sets, order effects and social desirability could bias the results, particularly concerning negative traits and in the case of religious participants with an idealized self-perception. This possibility should be addressed in the discussion section. Future studies on the validity of the SenPQ would benefit from the inclusion of social desirability measures and question randomization.

---

## Round 0.2 · accepted · Accept

Reviewer 2 has reviewed your revised manuscript and they are happy with the excellent job that you did.

·

Basic reporting

In their revised manuscript, the authors have properly acknowledged the reviewers' comments, and the paper was significantly improved. In particular, the introduction section now includes a detailed justification for the selection of the additional measures. The authors also added more data in the results, and expanded the discussion with important commentaries on the limitations of the investigation and perspectives for future studies. The paper is well-written and the results are relevant. I suggest the publication of the paper in its present form.

Experimental design

The research question is well-defined and meaningful. The paper gives an important contribution to the study of sensed presence experiences with the development of an original measure of this construct. The investigation was rigorously performed, and the methods section provides sufficient information. All appropriate raw data was made available and can be checked by other researchers.

Validity of the findings

The statistical analyses are sound and the results are robust. Conclusions are well-stated and the discussion section provides a detailed description of the study’s limitations , as well as perspectives for future research. I encourage replication of the study’s findings.

Additional comments

Congratulations to the authors! They have done an excellent job.